# The Bio-Persistence of Reversible Inflammatory, Histological Changes and Metabolic Profile Alterations in Rat Livers after Silver/Gold Nanorod Administration

**DOI:** 10.3390/nano11102656

**Published:** 2021-10-09

**Authors:** Ying Liu, Hairuo Wen, Xiaochun Wu, Meiyu Wu, Lin Liu, Jiahui Wang, Guitao Huo, Jianjun Lyu, Liming Xie, Mo Dan

**Affiliations:** 1CAS Key Laboratory for Biomedical Effects of Nanomaterials and Nanosafety, NCNST-NIFDC Joint Laboratory for Measurement and Evaluation of Nanomaterials in Medical Applications, Center for Excellence in Nanoscience, National Center for Nanoscience and Technology, No. 11 Beiyitiao Zhongguancun, Haidian District, Beijing 100190, China; liuy1@nanoctr.cn; 2National Center for Safety Evaluation of Drugs, National Institutes for Food and Drug Control, No. 8 Hongda Mid-Road, Beijing Economic and Technological Development Zone, Daxing District, Beijing 100176, China; hairuowen@163.com (H.W.); jhwang18@mails.jlu.edu.cn (J.W.); huoguitao@nifdc.org.cn (G.H.); or jjlv@innostar.cn (J.L.); 3CAS Key Laboratory of Standardization and Measurement for Nanotechnology, NCNST-NIFDC Joint Laboratory for Measurement and Evaluation of Nanomaterials in Medical Applications, Center for Excellence in Nanoscience, National Center for Nanoscience and Technology, No. 11 Beiyitiao Zhongguancun, Haidian District, Beijing 100190, China; wuxc@nanoctr.cn (X.W.); wumy@nanoctr.cn (M.W.); liulin@nanoctr.cn (L.L.); 4Department of Pathology, InnoStar Bio-Tech Nantong Co., Ltd., Nantong 226133, China; 5School of Nanoscience and Technology, University of Chinese Academy of Sciences, Beijing 100049, China; 6The State Key Laboratory of New Pharmaceutical Preparations and Excipients, 226 Huanghe Road, Shijiazhuang 050035, China

**Keywords:** silver nanomaterials, metabolic changes, long-term toxicity, recovery, liver toxicity, bio-persistence

## Abstract

As a widely applied nanomaterial, silver nanomaterials (AgNMs) have increased public concern about their potential adverse biological effects. However, there are few related researches on the long-term toxicity, especially on the reversibility of AgNMs in vivo. In the current study, this issue was tackled by exploring liver damage after an intravenous injection of silver nanorods with golden cores (Au@AgNRs) and its potential recovery in a relatively long term (8 w). After the administration of Au@AgNRs into rats, Ag was found to be rapidly cleared from blood within 10 min and mainly accumulated in liver as well as spleen until 8 w. All detected parameters almost displayed a two-stage response to Au@AgNRs administration, including biological markers, histological changes and metabolic variations. For the short-term (2 w) responses, some toxicological parameters (hematological changes, cytokines, liver damages etc.) significantly changed compared to control and AuNRs group. However, after a 6-week recovery, all abovementioned changes mostly returned to the normal levels in the Au@AgNRs group. These indicated that after a lengthy period, acute bioeffects elicited by AgNMs could be followed by the adaptive recovery, which will provide a novel and valuable toxicity mechanism of AgNMs for potential biomedical applications of AgNMs.

## 1. Introduction

Due to the broad antimicrobial spectrum and the high efficacy against specific bacteria, silver nanomaterials (AgNMs) have become the dominate nanomaterials widely applied in consumer products such as textiles, personal care, cosmetics, home furnishing, appliances, biomedicine, etc. [1,2]. There are approximately 55 tons products containing AgNMs commercially manufactured annually [3]. Therefore, a large amount of AgNMs have been unwittingly entering the environment and human bodies, which inevitably increases public concern about their potential adverse biological effects [4].

It is critical to understand the potential toxicity and reversibility of AgNMs in organs. Substantial in vitro experimental evidence has indicated that silver nanoparticles (AgNPs) may induce cytotoxicity through (1) an increase in oxidative stress; (2) the interaction with proteins by binding to free thiol groups; (3) ion-regulatory disturbances by the mimicry of endogenous ions (e.g., sodium, potassium, or calcium); and (4) integration with mRNA or microRNA [5,6,7,8,9]. However, some in vivo studies demonstrated the kinetics and tissue distribution pattern and toxicity of AgNPs generated in different animal models using different routes of exposure [10,11,12,13,14,15]. Most researchers consider that AgNPs can be rapidly cleared from blood and then transfer to various tissues quickly, followed by a distribution equilibrium [10,16,17]. The general consensus of these in vivo studies reveals that after i.v. injection of AgNPs, the highest concentration of Ag was found in the liver or spleen tissue [10,17,18,19]. As for the toxicity caused by AgNMs, the relative prevalence and severity of lesions are actually controversial probably as a result of the high variability of AgNMs in terms of size, coating, concentration and source. AgNPs have been reported to lead to midzone hepatocellular necrosis, splenic hyperemia, gall bladder hemorrhage, etc. [19]. However, there is few literature in respect of ultrastructural details in these damaged tissues. More importantly, there is few related research focusing on whether the AgNMs-induced toxicities are reversible or not in vivo, especially in the long-term toxicity study [20,21,22,23,24,25]. In addition, as the final products of gene expression, the profile changes of metabolites with low molecular weight are increasingly proved to be valuable for directly unveiling unforeseen biological effects [26,27]. However, the metabolic profile alterations elicited by AgNMs are rarely involved in many publications.

In the current study, we aimed to explore the liver damage and potential recovery caused by AgNMs by combining metabolomic analysis and classic toxicity assays, and the histological and ultrastructural changes in liver tissue were also examined. One type of special AgNMs, gold nanorods (AuNRs) with silver coating (Au@Ag NRs), was used in this study. It would be easier to compare total bio-effects of gold nanomaterials with those of AgNMs. In order to avoid absorption from the intestinal tract and first pass elimination by the liver, the intravenous (i.v.) administration route was chosen here to examine tissue distribution and elimination of AgNMs [28]. The time-dependent contents of Ag and Au in blood and their distribution in most organs of rats at different time points were analyzed after a single i.v. injection of AuNRs or Au@Ag NRs. Additionally, the metabolic profile alterations in liver addressed that Au@Ag NRs exposure had an influence on important metabolism pathways. Both histopathological and metabolic analysis suggested that the short-term toxicity caused by AgNMs would be repaired spontaneously in a long-term period. Therefore, our results will provide a great deal of qualitative and quantitative information on a wide range of metabolites correlated with Au@Ag NRs administration. Moreover, a novel self-repair hypothesis was addressed to better understand the long-term bio-effects of AgNMs in their various applications.

## 2. Materials and Methods

### 2.1. Materials

Chlorauric acid (HAuCl_4_·3H_2_O), sulphuric acid (H_2_SO_4_), sodium chloride (NaCl), and silver nitrate (AgNO_3_) were purchased from Beijing Chemical Reagent Company (Beijing, China). L-ascorbic acid (AA) and sodium borohydride (NaBH_4_) were from Alfa Aesar (Haverhill, MA, USA). Poly (diallyldimethylammonium chloride (PDDAC, M.W. 100,000–200,000), poly (sodium p-styrensulfonate) (PSS, M.W. 70,000) and cetyltrimethylammonium bromide (CTAB) were obtained from Amresco (Solon, OH, USA). Sucrose was purchased from Sinopharm Chemical Reagent Co., Ltd. (Beijing, China). Deionized water (18 MΩ·cm) was used in this study.

### 2.2. Preparation and Characterization of Gold Nanorods (AuNRs) and Gold Core/Silver Shell Nanorods (Au@Ag NRs)

AuNRs was synthesized via a modified seed-mediated method [29]. Chemical reduction of HAuCl_4_ by NaBH_4_ provided CTAB-capped Au seeds. A volume of 7.5 mL of CTAB solution (0.1 M) was mixed with 100 μL of 24 mM HAuCl_4_ and diluted with water to 9.4 mL. Then, 0.6 mL of ice-cold NaBH_4_ (0.01 M) was added with magnetic stirring for 3 min. After that, the seed suspension was kept at 30 °C for 2 h. Aging of seeds (2–5 h) was found to give high rod yield of AuNRs.

The growth solution of the AuNRs was prepared by mixing CTAB (0.1 M, 100 mL), HAuCl_4_ (25.5 mM, 1.96 mL), AgNO_3_ (0.01 M, 1.2 mL), H_2_SO_4_ (0.5 M, 2 mL), and AA (0.1 M, 0.8 mL), followed by the addition of 100 μL of seed solution to initiate the AuNRs growth. Next, 0.4 mL of AA (0.1 M) and 0.98 mL of HAuCl_4_ (25.5 mM) were added after 55 and 65 min, respectively. After 12 h, AuNRs with longitudinal surface plasmon resonance (LSPR) at ~870 nm were obtained and purified twice by centrifugation (12,000 rpm for 10 min). The precipitates were redispersed in 200 mL deionized water.

Ag-shelled AuNRs were prepared by heat-assisted Ag overgrowth in the presence of CTAB [30]. The growth solution consisting of CTAB (0.1 M, 10 mL), AgNO_3_ (0.1 M, 1.5 mL) and AA (0.1 M, 15 mL) was added into 100 mL above purified AuNRs suspension. The mixed solution was kept stirring at 70 °C for 8 h to let Ag shell grow.

Ag-shelled AuNRs were centrifuged at 9000 rpm for 7 min to remove excess CTAB and re-suspended with the same volume of deionized water.

### 2.3. Surface Coating of AuNRs and Au@Ag NRs with PDDAC

Typically, 1 mL above-prepared CTAB-coated NRs suspension was added into 50 mL aqueous solution containing 20 mg/L PSS and 6 mM NaCl. After 12 h, the solution was centrifuged at 9500 rpm for 10 min to remove excess PSS and the precipitate was dispersed by 1 mL deionized water. Similarly, 1 mL PSS-coated NRs was added into 50 mL aqueous solution containing 20 mg/L PDDAC and 6 mM NaCl to obtain PDDAC-coated NRs.

The morphology and size of AuNRs as well as Au@Ag NRs were characterized by transmission electron microscopy (TEM, HT7700, Hitachi, Tokyo, Japan). A total of 270 AuNRs and 227 Au@Ag NRs were counted, respectively. The UV-Vis spectrum was measured by UV/visible/near infrared (UV/Vis/NIR) spectrophotometer (Lambda 950, Perkin Elmer Instruments Co. Ltd., USA) from 200 nm to 1100 nm. Zeta potential was tested by dynamic light scattering (DLS) (Zetasizer Nano ZS, Malvern, UK) in a clear disposable zeta cell. The contents of Au and Ag in nanorods were measured by inductively coupled plasma mass spectrometry (ICP-MS) system (NexION 300X, Perkin Elmer, Waltham, MA, USA).

### 2.4. Animal Experiment and Sample Collection

All the protocols of animal experiments and sample collections were performed in the National Center for Safety Evaluation of Drugs (NCSED) and approved by the Institutional Animal Care and Use Committee of NCSED in compliance with China’s national ethical standards to minimize the suffering of animals (IACUC- 2017-K017).

Wild-type specific-pathogen free male Sprague Dawley (SD) rats of 6 weeks old (with body weights between 174 g and 218 g) were purchased from Beijing Vital River Laboratory Animal Technology Co., Ltd. (Beijing, China; Animal Quality Certificate No. SCXK [Jing] 2016-0006). After a one-week acclimatization to laboratory conditions, rats were randomly divided into three groups (*n* = 8 for each), media control (5% sucrose solution), AuNRs, and Au@Ag NRs. Each group was further evenly divided into two groups, 2-week (2 w) and 8-week (8 w) recovery groups (*n* = 4 per group). A total of 5 mg/kg Au@Ag NRs was chosen in our study according to our previous data [31,32]. As the Au content in 5 mg Au@Ag NRs was estimated to be 4 mg based on the ration of Ag/Au (Ag/Au = 1.2), the content of Au administered to rats was the same in the nanomaterial-treated groups. Then, 5% sucrose solution, 4 mg/kg AuNRs, and 5 mg/kg Au@Ag NRs in the volume of 5 mL/kg were intravenously (i.v.) injected into the corresponding groups of rats by a single dose, respectively. All animals were kept for 2 or 8 weeks for the tissue collection.

Body weight and food uptake of rats were monitored once per week for the 8 w group after administration. At the indicated time points after intravenous exposure (0 min, 1 h, 6 h, 48 h, 2 w for each 2 w group (*n* = 4); 10 min, 3 h, 24 h, and 1, 3, 4, 6, and 8 w for each 8 w group (*n* = 4)), blood samples were collected for further experiments. Furthermore, all rats of 2 w and 8 w groups (*n* = 4 per group) were sacrificed for tissue collection at the time points of 2 w and 8 w, respectively.

### 2.5. Hematological Examination

The blood samples (0.5 mL) collected at 1, 2, 4 and 8 w in EDTA-tubes were used to detect hematological parameters including the number of white blood cells (WBC), neutrophils (NEUT), lymphocytes (LYMPH), monocytes (MONO), eosinophils (EOS), basophils (BASO), red blood cells (RBC) and reticulocyte (#Retic), the percentage of neutrophils (%NEUT), lymphocytes (LYM%), monocytes (%MONO), eosinophils (%EOS), basophils (%BASO) and reticulocytes (%Retic), hemoglobin (HGB), hematocrit (HCT), mean corpuscular volume (MCV), mean corpuscular hemoglobin (MCH), mean corpuscular hemoglobin concentration (MCHC), platelet count (PLT), and mean platelet volume (MPV).

### 2.6. Biochemical Assay of Serum

The serum samples were examined by Hitachi 7180 Biochemistry Automatic Analyzer (Hitachi Ltd., Gyeonggi-do, South Korea) for classic indexes includes: alanine aminotransferase (ALT), aspartate aminotransferase (AST), alkaline phosphatase (ALP), creatine kinase (CK), lactate dehydrogenase (LDH), total bilirubin (TBIL), urea (UREA), creatinine (CRE), glucose (GLU), total cholesterol (CHO), triglyceride (TG), total protein (TP), albumin (ALB), ALB/GLB (A/G), serum potassium (K^+^), serum sodium (Na^+^), serum chloride (Cl^−^).

### 2.7. Histopathology

All liver tissues were routinely fixed in a 10% formalin solution and embedded in paraffin, cut into 3–5 μm sections and stained with hematoxylin and eosin (H&E) for histopathologic examination by light microscopy.

### 2.8. The Distribution of AuNRs or Au@AgNRs in Liver Tissues by TEM

Liver tissues were collected after AuNRs or Au@AgNRs exposure were fixed in 2% formaldehyde/2.5% glutaraldehyde and then in 1% osmium tetrosxide. After washed with the sodium cacodylate buffer and dehydrated by gradient ethanol, tissues were dipped into propylene oxide and embedded in Epon 812 (Beijing Xinxingbairu Co., Ltd., Beijing, China). Tissue sections (1 μm) were cut, stained with uranyl acetate and lead citrate, and examined under an H-7650 electron microscope. (HITACHI, Ibaraki, Japan).

### 2.9. Cytokine Detection in Serum

The blood samples harvested at 6 h, 1 w, 2 w, 4 w and 8 w were kept at 4°C for 30 min, followed by centrifugation at 6000 rpm for 5 min. The supernatant was collected as serum samples and examined for cytokine concentrations by Milliplex Catalog ID. RECYTMAG-65K-20. Rat Expanded Cytokine Magnetic (# RECYTMAG- 65K-02, Millipore).

### 2.10. Metabolite Analysis in Liver Tissue Extracts

The metabolomics profiling of liver tissues was performed by XploreMET platform (Metabo-Profile, Shanghai, China). The liver samples were prepared according to the previously published method with minor modifications [33,34].

### 2.11. Statistical Analysis

All data in this study were presented as mean ± standard deviation (SD). Inter-group variations were calculated by two-way ANOVA. *p* values of less than 0.05 (*p* < 0.05) was considered as statistically significant.

## 3. Results

### 3.1. Physicochemical Characterization of AuNRs and Au@Ag NRs

The gold nanorods (AuNRs) as a control nanomaterial had an Au concentration of 714 ± 5 mg/L. The Ag-shelled AuNRs (Au@Ag NRs) were synthesized using AuNRs as templates with the Ag/Au atomic feed ratio of ~2.3 (Figure 1a). The content of Ag and Au in Au@Ag NRs was 678 ± 10 and 534 ± 3 mg/L, respectively, measured by ICP-MS. EDX imaging demonstrated the presence of Ag on the surface of AuNRs by EDX analysis from our previously published data [10]. UV-Vis absorption spectra showed the longitudinal surface plasmon maximum was 868 nm and 682 nm for AuNRs and Au@Ag NRs dispersed in water, respectively (Figure 1b). TEM imaging in Figure 1c showed that the average length and diameter of AuNRs were 66.7 ± 10.0 nm and 15.0 ± 2.5 nm, and the average aspect ratio (AR) was 4.5 ± 0.8. As well, Figure 1d gave the size and morphology information of Au@Ag NRs. Owing to preferred lateral growth, Au@Ag NRs showed decreased AR to 2.8 ± 0.4 with the average of diameter of 26.2 ± 3.0 nm and the length of 72.7 ± 8.9 nm, respectively. The corresponding thickness of the Ag shell was ~3 nm along the longitudinal axis of Au@Ag NRs and ~5.6 nm along the transverse axis of Au@Ag NRs, respectively. It can be also found that the surface of AuNRs was covered by a homogeneous layer of Ag with a thickness of ~5.0 nm. Zeta potential of these two kinds of nanorods in water was 37.7 ± 1.6 mV (AuNRs) and 52.5 ± 1.4 mV (Au@Ag NRs), respectively.

### 3.2. The Contents of Au and Ag in Blood and Organs after the Exposure of Au@Ag NRs

According to the surveillance data from each week under our experimental conditions, there were no significant differences of body weight and food uptake between negative control and NRs-treated groups (Appendix A). Due to the dependence on the control of their distribution within body for the efficacy of nanomaterial applications, it is important to demonstrate the concentration–time curve in blood and organs of interest. In all tissue samples, Au or Ag concentration was measured by ICP-MS. As shown in Figure 2a, all concentration–time curves revealed a rapid decline in metal concentration during the first 10 min after injection. After 10 min, blood concentrations of Au or Ag remained stable until 8 w and rather low, quite close to the blank value. It was noticed that Ag concentration in blood at the beginning (0 min) was higher than Au after AuNRs or Au@Ag NRs. Moreover, compared to AuNRs, the blood content of Au from 0 min to 10 min was a little bit more than that after Au@Ag NRs exposure.

Major organs were harvested at 2 w and 8 w after a single i.v. injection of AuNRs or Au@Ag NRs. Figure 2b–d provided the whole data from all tested organs. Both at 2 w and 8 w after the administration of AuNRs (Figure 2b), there were comparatively high levels of Au in liver and spleen, followed by lung, which was similar to the Au profile of Au@Ag NRs (Figure 2c). The only difference was the Au concentration pattern of lung between at 2 w and 8 w. At 8 w after AuNRs injection, the Au level was remarkably lower than that at 2 w in the lung. However, there was similar amount of Au in lung both at 2 w and 8 w after Au@Ag NRs exposure. In addition, as shown in Figure 2d, Ag was mainly distributed in liver and spleen, which was much less than Au in Au@Ag NRs-treated organs at 2 w or 8 w. In most tested organs, the Ag levels at 8 w were lower than those at 2 w. However, only Ag in the livers of Au@Ag NRs-treated rats still remained the relatively high level at 8 w.

### 3.3. Hematological Changes after the Exposure of Au@Ag NRs

We detected the hematological indicators in the whole blood samples at 1, 2, 4, and 8 w (Appendix A). At 1 w, only total number (Retic) and percentage of reticulocytes markedly raised after the injection of AuNRs or Au@Ag NRs. At 2 w, Au@Ag NRs aroused the increased percentage of neutrophils (NEUT) along with the reduced number of monocytes compared to control or AuNRs group. While at 4 and 8 w, there were no significant changes for these tested hematological indicators after the treatment of Au@Ag NRs.

### 3.4. Effects of Au@Ag NRs on Serum Biochemical Markers

Generally, in clinical examination, the biochemical markers can be used to evaluate organ functions especially for kidney and liver, and disease prognoses. In this study, we carried out serum biochemical assay to determine the effects of Au@Ag NRs at 2 w or 8 w after exposure. As shown in Appendix A, the concentration of TG increased to almost 192% in the Au@Ag NRs-injected group at 2 w. However, at 8 w after exposure of Au@Ag NRs, we cannot see any effects on all detected biochemical markers.

### 3.5. The Release of Cytokine or Grow Factor in Serum

To explore the inflammation caused by nanorods in rats, we measured the release of five Th1 type and two Th2 type cytokines as well as one growth factor in serum after nanorod exposure. After a short-time period (6 h), the concentration of all tested cytokines in serum did not markedly change (Figure 3). As the members of interleukin (IL)-1 family, IL-1α and IL-1β displayed the different change patterns (Figure 3a,b). Compared to the control or AuNRs group, after Au@Ag NRs administration, IL-1α release increased from 1 w, reached peak at 2 w, and then followed by declining to the threshold at 8 w, which was similar to interleukin 6 (IL-6) (Figure 3c) as well as interleukin 12 (IL-12p70) (Figure 3d). In contrast, Au@Ag NRs exposure made the concentration of IL-1β peak at 4 w and still remain a relatively high level at 8 w in serum (Figure 3b), which was similar to interleukin 2 (IL-2) (Figure 3e). As one of the most prominent inflammatory cytokines, tumor necrosis factor (TNF-α) showed the highest level at 2 w after the injection of Au@Ag NRs (Figure 3f). As for Th2 cytokines, interleukin 4 (IL-4) and interleukin 10 (IL-10) displayed the similar change profile to IL-1β (Figure 3g,h). In addition, we also checked vascular endothelial growth factor (VEGF) in serum. As shown in Figure 3i, Au@Ag NRs provoked the increase of VEGF since 1 w and peaked at 2 w, similar to IL-1α.

### 3.6. Effects of Au@Ag NRs on Histological Changes

Harvested liver tissues were conducted by histopathological examinations after 2-week or 8-week exposure. As shown in Figure 4a–c, Au@Ag NRs as well as AuNRs could induce apparent inflammatory foci near the blood vessel in the liver parenchyma at 2 w. However, at 8 w, only small inflammatory foci were observed in the control group (Figure 4d), whereas we did not detect any histopathological changes in the liver tissues from all three groups (Figure 4d–f). Then, TEM was used to explore the ultrastructure variations in the liver tissue. At 2 w after nanorod exposure, some pyknotic nuclei obviously appeared in liver parenchyma compared with the normal nuclei of control liver (Figure 4g,h). Additionally, Au@Ag NRs injection led to distinct vacuolization and apoptosis characteristic of nuclear fragmentation in some hepatocytes (Figure 4i). Moreover, the EDX mapping of TEM imaging also confirmed the existence of both AuNRs and Au@Ag NRs inside liver cells at 2 w after administration (Figure 5).

### 3.7. Metabolic Responses of Liver Tissues to Au@Ag NRs Treatments

A total of 229 metabolites were identified and analyzed based on both retention times and fragmentation patterns in the mass spectrum. Partial least-squares discriminate analysis (PLS-DA), a supervised clustering method, provides the maximized separation between groups. As shown in Figure 6a,b, the score plot of PLS-DA multivariate analysis showed that only Au@Ag NRs group was clearly separated from the control, especially after 2 w exposure. However, AuNRs group was similar to the control at 2 w or 8 w. Additionally, as shown in Appendix A, it was clear that there were more regulated metabolites caused by the nanorod administration at 2 w than at 8 w.

In order to discriminate AuNRs and Au@Ag NRs responses at 2 w, the metabolic data sets of Au@Ag NRs versus AuNRs by orthogonal PLS-DA (OPLS-DA) analysis on basis of a multi-criteria assessment and univariate statistical analysis (*t* test) were shown in Table 1. More than 30 strikingly regulated metabolites based on both multivariate and univariate results were listed in Table 1. These metabolites involved alcohols (glycerol), amino acid (pyroglutamic acid, L-threonine, L-leucine, L-asparagine, gamma-aminobutyric acid), carbohydrates (L-arabinose, cellobiose, kojibiose, mannitol, isomaltose, alpha-lactose, D-maltose, L-arabitol and ribitol), fatty acids (tetracosanoic acid, arachidonic acid, docosahexaenoic acid, heptadecanoic acid, myristic acid, palmitic acid, palmitoleic acid and oleic acid), lipids (MG160 and MG182), nucleotide (uridine), organic acids (orotic acid and petroselinic acid) and vitamin (pantothenic acid and Alpha-tocopherol).

As shown in Table 1, compared to control group, few metabolites were significantly changed after AuNRs administration, while numerous metabolites varied with Au@Ag NRs injection. That being said, most of metabolite variations were Au@Ag NRs specific, which was further supposed to be AgNMs specific. Only the level of uridine was significantly elevated both by AuNRs and by Au@Ag NRs. Moreover, the elevation of uridine in the Au@Ag NRs group was ~2.6 fold to that in the AuNRs group. In addition, Figure 6c–f showed that the changes of some important metabolites, including glycerol, L-leucine, L-asparagine and uridine. A similar tendency was observed in these metabolites. Consistently, Au@Ag NRs led to more significant changes at 2 w than those at 8 w.

In addition, the Au@Ag NRs-specific changes of some metabolites were also correlated with cytokine variations. Glycerol, L-leucine and uridine were positively correlated with some important cytokines including IL-1α, IL-1β, IL-6, IL-12p70, L-2, IL-4, IL-10, as well as VEGF (Appendix A).

### 3.8. The Metabolic Networks Related to Au@Ag NRs via the Metabolite Variations in the Liver Tissue

Based on more change tendency of metabolites at 2 w, a global analysis of potential biomarkers can highlight the key metabolic pathway network to provide an overall impression of biological effects induced by Au@Ag NRs. Figure 7 displays the disturbed metabolic pathways were mainly some common metabolic responses including amino acid metabolism, fatty acid metabolism and energy metabolism (glycolysis). At 2 w, livers of Au@Ag NRs treated rats showed elevated glucose 6-phosphate (G6P) and decreased glucose, which were correlated to the process of glycolysis and tricarboxylic acid (TCA) cycle. Along with upregulated uridine, glycerol, a product of triglycerides metabolism in liver, was also increased. Meanwhile, upregulated L-leucine can also mediate hepatic lipid metabolism and associated changes in lipogenic gene expression (fatty acid synthase, stearoyl CoA desaturase, acetyl CoA carboxylase). The integrated metabolic networks derived from the liver metabonomic variations revealed another notable effect upon Au@Ag NRs exposure at 2 w. The decreased levels of L-glutamine as well as GSH were involved in the D-Glutamine and D-glutamate metabolism (Figure 7).

## 4. Discussion

The increasing number of investigations reported the bio-toxicity of AgNMs in vitro and in vivo and resulted in negative impacts on their applications. few studies have revealed the long-term toxicities and ultrastructural changes as well as metabolism variations induced by AgNMs, and thus obtaining more in-depth information about these important aspects in biological systems will be helpful for their toxicology and risk assessment. In this study, the biological effects of a special silver nanomaterial (Au@Ag NRs) intravenously administered to rats were comprehensively assessed. Here, the dosage of 5 mg/kg Au@Ag NRs was chosen according to our previous data. An amount of 5 mg/kg AgNPs can elicit severe abnormal clinical symptoms by a single i.v. injection [32]. In HepaRG cells, Au@Ag NRs were supposed to be more toxic than AgNPs [31]. In addition, 5 mg/kg was estimated as the maximum tolerated dose of Au@Ag NRs by i.v. injection in rats based on our unpublished data. Additionally, CTAB/PSS was used for the preparation of Au@Ag NRs in our study since CTAB is usually used as the stabilizing agent during preparation of AuNRs [35]. The synthesis method of Au@Ag NRs was the same as that in the previous paper by Meng J et al. [36]. Although CTAB is a toxic cationic surfactant [37], it has been confirmed that CTAB could not significantly cause toxic effects in Meng’s paper. Moreover, other published papers have also verified that CTAB-coated AuNRs have low toxicity in mice after two weeks by an intravenous injection [38]. Hence, based on these data, CTAB in the Au@Ag NRs should not cause potential toxicity in our study.

Under our experimental conditions, the NRs-treated rats showed normal behaviors and no evidence of gross toxicity as well as similar body weight and food uptake, comparing to the control group (Appendix A). Based on the hematological analysis (Appendix A), a transient increase of reticulocytes (Retic) at 1 w possibly reflected the potential hemolytic effects induced by NRs, especially AuNRs, which was supported by some previous reports about gold nanoparticles-elicited hemolysis [39,40]. Only the slight increase of neutrophil number accompanied with the decreased number of lymphocytes at 2 w suggested that there were inflammatory responses within rat bodies after NRs treatment, especially in the Au@Ag NRs-treated group. However, all other parameters were similar to control rats. Moreover, no striking changes could be observed between control and treated groups after 2 w (e.g., 4 w or 8 w). It meant that the inflammatory responses induced by NRs may slowly disappear after a long term. In addition, the analysis of serum biochemistry further showed elevated levels of TG (~2 folds) as a biomarker for lipid metabolism in the Au@Ag NRs-treated group at 2 w (Appendix A) [41]. Similarly, at 8 w, TG reduced to the normal level compared to the negative control group, which suggested that Au@Ag NRs may cause lipid metabolism disorders in a short-term period (e.g., ~2 w), whereas these effects would disappear after a long time (~8 w).

There were the relatively high levels of Ag in liver and spleen at both 2 w and 8 w. This result was compatible with the distribution patterns of AgNPs published in other previous papers [36,42,43,44]. Moreover, the EDX mapping of TEM imaging also confirmed the existence of NRs inside liver cells at 2 w after administration (Figure 5). It was noted that at 8 w after administration, only Ag in the liver of Au@Ag NRs-treated rats did not decline and still remained the relatively high level. It confirmed that the liver may be the leading target organ for nanoparticle accumulation, especially for AgNMs during a long-term period, although thus far there were no related referrable publications [45]. As a result of such a long-term accumulation of AgNMs in liver, the bio-effects of AgNMs for liver functions must not be neglected. Therefore, the histological examination was further conducted on the liver tissues at 2 w or 8 w. At 2 w after administration, some pyknotic nuclei obviously appeared in liver parenchyma of both AuNRs and Au@Ag NRs treated groups, and also Au@Ag NRs injection exclusively led to distinct vacuolization and apoptosis characteristic of nuclear fragmentation in some hepatocytes (Figure 4i). These results indicated that AuNRs elicited shrunken and dark nuclei, while Au@Ag NRs resulted in apoptosis. It has been reported that AgNPs can lead to apoptosis in hepatocytes in a number of publications [46,47]. It has been also reported that upon AgNPs exposure for several hours, livers exhibited mild degenerative changes including the cytoplasmic vacuolization of hepatocyte [48]. Moreover, this type of cytoplasmic vacuolization without the filling of lipid and glycogen could be a beneficial adaptation of hepatocytes towards a toxic insult [49]. However, at 8 w, there were no obvious histological changes in the liver tissue, which suggested that those observable lesions induced by Au@Ag NRs may be transient and can be recovered, which may not lead to persistent liver injury. These results hinted that the AgNMs-elicited bio-effects in the liver were reversible and repairable, which was consistent with the data of hematological analysis as well as serum biochemical assay. The lack of striking changes in blood parameters and histological features in a long-term administration in our present study also verified that AgNMs at doses <10 mg/kg bw could be well tolerated when intravenously administrated into mice or rats, which was consistent with previous research on the relative short-term studies [16,44,48].

Although the expression levels of all the tested cytokines were elevated by Au@Ag NRs at 2 w, among all the tested pro-inflammatory Th1 type cytokines, only IL-1β and IL-2 still remained relatively higher levels at 8 w. As previously reported, AgNPs induced inflammasome and significantly high levels of IL-1β production in Hep G2 cells, a human hepatoma cell line [50]. Recently, it has been confirmed that an imbalanced expression of IL-1β could mediate liver injury [51]. AgNPs can also lead to the significant increase of IL-1β in primary rat brain microvessel endothelial cells (rBMEC) [52]. In addition, the instillation of 20 nm gold-core AgNPs elicited a moderate, and non-statistically significant increase of IL-2 [53]. This indicated that in our case, at 8 w after administration of Au@Ag NRs, there may be still slight or unobvious changes in the liver tissue, which were not repaired yet. In addition, our results also showed that, at 8 w, Th2 type cytokines (IL-4 and IL-10) still remained in the relatively higher levels. AgNPs have been reported as a potential adjuvant to elicit the Th2-biased immune responses by increasing IgG1/IgG2a ratios in mice [54]. Moreover, it has been demonstrated that the secretion of Th2 type cytokines was more predominant than that of Th1 type cytokines on day 28 after a single instillation [55]. Thus, it was supposed that in a short term (~2 w) AgNMs would dominantly induce Th1 type inflammation, while in a long term (~8 w) AgNMs may preferably cause slight Th2 type immune responses. This hypothesis still further needed a vast amount of evidence. Moreover, this study addressed the correlation between some changed metabolites and cytokines (Appendix A). Since important metabolites have been confirmed to play pro-inflammatory roles during inflammation, our results may provide further proof for AgNMs -induced inflammation [56].

Consistent with the abovementioned conventional toxicity readouts, the metabolic variations related to carbohydrate, amino acid and lipids metabolism in liver also clearly indicated a two-stage response during the administration of Au@Ag NRs. At the short-term stage after injection (2 w), the elevated G6P indicated Au@Ag NRs may induce glycolysis and impact TCA cycle in the liver, suggesting a metabolic shift toward glycogenesis [57]. It has been reported that as a short-term (several hours) response of AgNPs, glycogen depletion as well as lipid synthesis/storage in the liver of treated mice possibly suggested that AgNPs impaired the glycogen synthesis and lipid catabolism and or transport [48,57]. However, in our case, the long-term (several weeks) responses of Au@Ag NRs involved neither significant changes in metabolites nor obvious fat droplets in liver, which implicated that the hepatic histopathological changes together with metabolic variations observed at 2 w were probably transient and may not cause persistent and severe liver injuries [48]. Moreover, along with upregulated uridine which prevents fatty liver [58]. glycerol, a product of triglyceride metabolism in liver, was also increased. Meanwhile, upregulated L-Leucine can also mediate hepatic lipid metabolism by adenosine 5‘-monophosphate (AMP)-activated protein kinase (AMPK) and associated changes in lipogenic gene expression (fatty acid synthase, sterol CoA desaturase, acetyl CoA carboxylase) in the liver [59]. It was supposed that the organism itself can upregulate lipolysis to initiate the prevention from lipid synthesis/storage in the liver at 2 w, which could be considered as a beneficial adaptation of hepatocytes towards a toxic insult [49]. After the 6-week recovery, amino acid, lipids and glucose metabolism recovered to normal levels compared with the control at 8 w. In addition, the lack of striking changes in blood parameters at 8 w after Au@Ag NRs administration further illustrated that AgNMs may be a type of well-tolerated nanomaterial, which was consistent with previous reports [16,44,48].

In our study, although the silver levels still remained high in major tissues at 8 w, unexpectedly, there was no obvious toxicity in Au@Ag NRs-treated rats, which was different from that at 2 w (Figure 2d). Therefore, it was supposed that there may be an adaptive mechanism for a long-term duration. A large amount of evidence has elucidated that AgNPs-elicited toxicity mainly results from oxidative stress which can reflect the imbalances in the cellular oxidative state [60,61]. Nel et al. proposed a hierarchical oxidative stress (HOS) model including a three-tiered, time-dependent cellular response to nanomaterials [62]. A previous report has revealed that bio-persistence of nanoceria can lead to a new phase (Tier-4) of oxidative stress in vivo, in which some markers of oxidative stress return to be normal [63]. Our data (Figure 3) confirmed that at 8 w after Au@Ag NRs administration some HOS-related inflammatory cytokines and metabolic profile changes returned to normalcy or tended to recover from higher levels. This indicated that after a lengthy period the toxicity elicited by Au@Ag NRs could be followed by the adaptive recovery which may aid organisms to a moderate or intermittent stress to attenuate detrimental bio-effects [64]. However, thus far, there are few reports evidencing on recovery responses after AgNMs-induced toxicity. Lee JY et al. published an in vivo study about AgNPs recovery in 2018 [65]. Although the materials as well as the administration doses are not exactly the same as our experimental system, Ag concentration in some main tissues (liver, kidney, spleen, lung and testis) after 4 w of AgNPs/AuNPs mixed administration decreased dose-dependently after another 4 w of recovery. However, it is a pity that this paper only provided the tissue distribution pattern but no other toxicity data after recovery. Therefore, our current work will be the first comprehensive in vivo study of AgNMs on the long-term toxicity followed by recovery. In the future, the adaptive mechanism for a long term will be investigated by the further exploration of clearing the residual Ag/Au remaining in some important organs (liver) to provide the key information for the bio-persistence of nanomaterials and their potential biomedical applications.

## 5. Conclusions

This study has demonstrated long-term bio-effects of AgNMs in the liver tissue by combining classic toxicity assays with metabolomic analysis. Especially, a novel insight into the metabolic adaptations undergone by liver tissues was provided in response to intravenously administered Au@Ag NRs at a dose which cannot elicit overt toxicity in the end of experiments. After the intravenous injection of Au@Ag NRs, Ag was rapidly cleared from blood within 10 min and mainly accumulated in the liver as well as spleen until 8 w. All detected parameters almost displayed a two-stage response to the administration of 5 mg/kg Au@Ag NRs, including blood markers, histological changes and metabolic variations. As the short-term (2 w) responses, the increased percentage of neutrophils (NEUT) along with the reduced number of monocytes showed inflammation responses in the Au@Ag NRs-treated rat blood. In the serum, Au@Ag NRs induced the remarkably higher levels of TG, which was related with lipid metabolism. Histologically, Au@Ag NRs injection led to distinct vacuolization and apoptosis characteristic of nuclear fragmentation in some hepatocytes at 2 w. Accordingly, more than 30 strikingly regulated metabolites (glycerol, L-leucine, uridine, etc.) were found based on both multivariate and univariate results in the metabolic analysis. Common metabolic pathways including amino acid metabolism, fatty acid metabolism and energy metabolism (glycolysis) were disturbed by Au@Ag NRs. However, after a 6-week recovery, all above-mentioned changes mostly returned to the normal levels. All these results suggest that AgNMs can be considered as a type of well-tolerated nanomaterial since the short-term bio-effects induced by AgNMs can be spontaneously recovered after a long-term restoration. It will provide new prospects about toxicity assessment for the future development of AgNMs applications. Moreover, the mechanism of adaptive recovery will be significantly helpful for the bio-persistence of nanomaterials and their potential biomedical applications.

## Figures and Tables

**Figure 1 nanomaterials-11-02656-f001:**
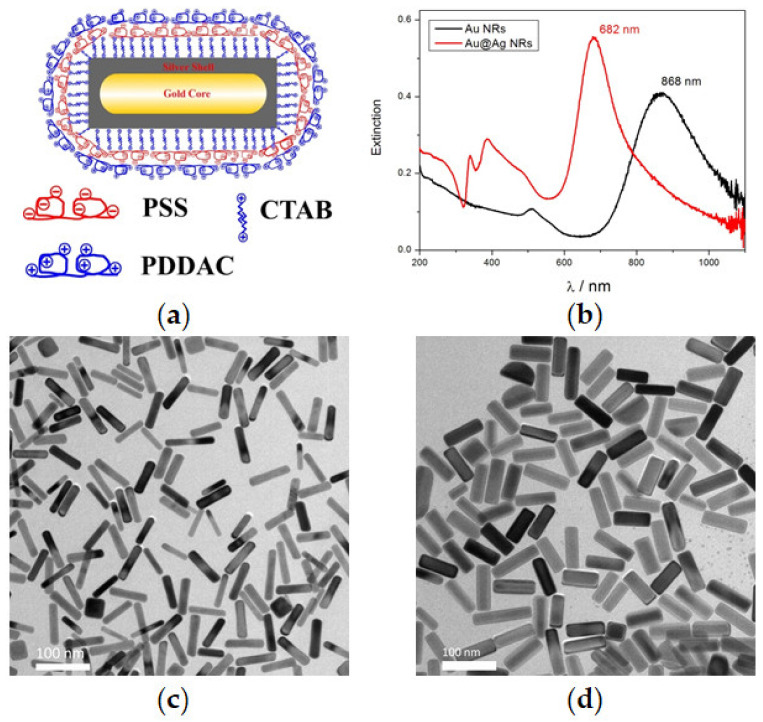
Physicochemical characterization of AuNRs and Au@Ag NRs used in animal experiments. (**a**) Schematic illustration of Au@Ag NRs. (**b**) UV-Vis spectrum. (**c**,**d**) Representative TEM images of AuNRs (**c**) and Au@Ag NRs (**d**), respectively.

**Figure 2 nanomaterials-11-02656-f002:**
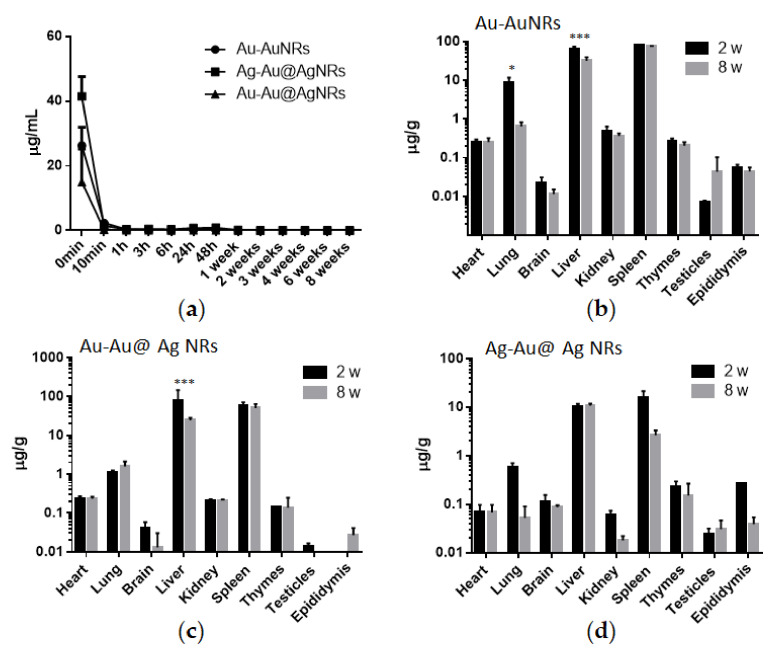
The contents of Au and Ag in tissues after AuNRs or Au@Ag NRs exposure. (**a**) The concentration–time curve in blood for Au or Ag. (**b**) The Au content in organs after the AuNRs exposure. (**c**,**d**) The content of Au or Ag after the Au@Ag NRs exposure. * *p* < 0.05; *** *p* < 0.001.

**Figure 3 nanomaterials-11-02656-f003:**
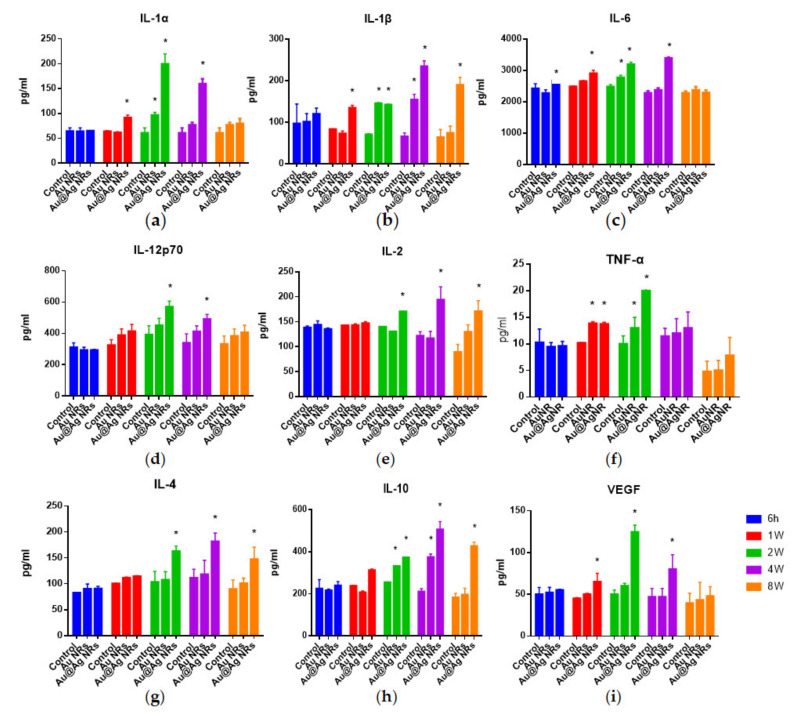
Cytokine and grow factor detection in serum. At 6 h, 1 w, 2 w, 4 w and 8 w after administration, the concentrations of IL-1α (**a**), IL-1β (**b**), IL-6 (**c**), IL-12p70 (**d**), IL-2 (**e**), TNF-α (**f**), IL-4 (**g**), IL-10 (**h**) and VEGF (**i**) were detected in serum, respectively. * *p* < 0.05.

**Figure 4 nanomaterials-11-02656-f004:**
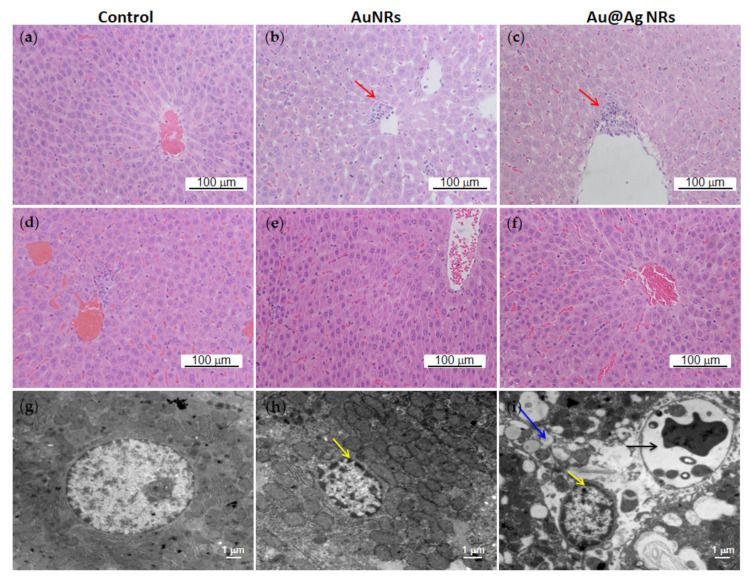
The changes of liver tissues. The liver tissues collected at 2 or 8 w after injection were sectioned. (**a**–**c**) Representative HE images at 2 w. (**d**–**f**) Representative HE images at 8 w. (**g**–**i**) Representative TEM images at 2 w. Red arrow, inflammatory foci. Yellow arrow, pyknotic nuclei. Blue arrow, cellular vacuolization. Black arrow, nuclear fragmentation.

**Figure 5 nanomaterials-11-02656-f005:**
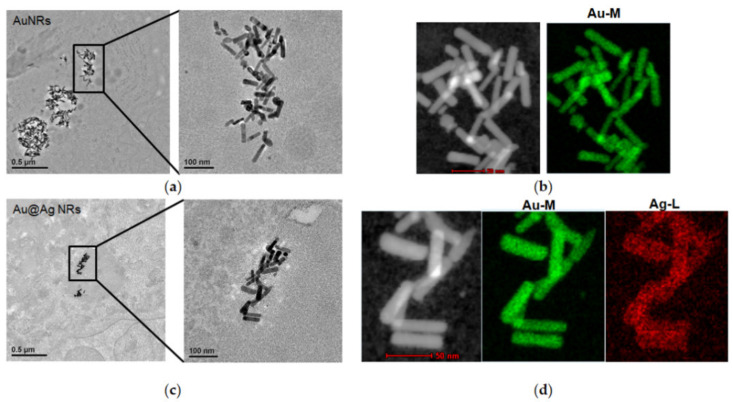
The existence of nanorods in hepatocytes. Representative TEM images as well as EDX mapping analysis for AuNRs (**a**,**b**) and Au@Ag NRs (**c**,**d**) in hepatocytes at 2 w after injection were shown.

**Figure 6 nanomaterials-11-02656-f006:**
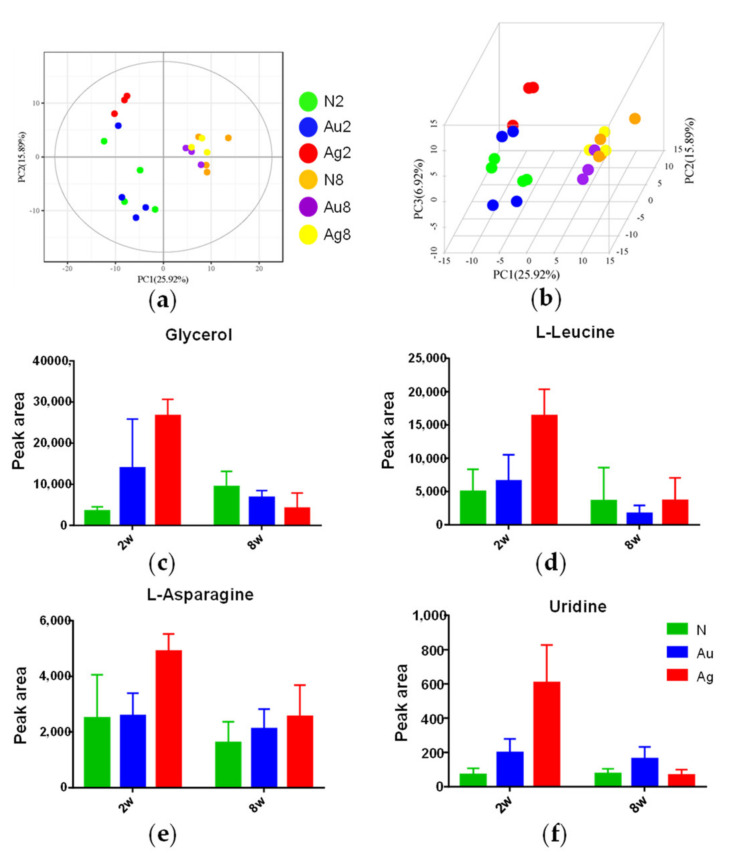
The PLA-DA score plots of metabolic profiles in liver tissues after the treatment of AuNRs or Au@Ag NRs. The score plot of PLS-DA multivariate analysis at (**a**) 2 w and (**b**) 8 w. (N2, the control group at 2 w; Au2, the AuNRs group at 2 w; Ag2, Au@Ag NRs at 2 w; N8, the control group at 8 w; Au8, the AuNRs group at 8 w; Ag8, Au@Ag NRs at 8 w). (**c**–**f**) The changes of some important metabolites at 2 w or 8 w.

**Figure 7 nanomaterials-11-02656-f007:**
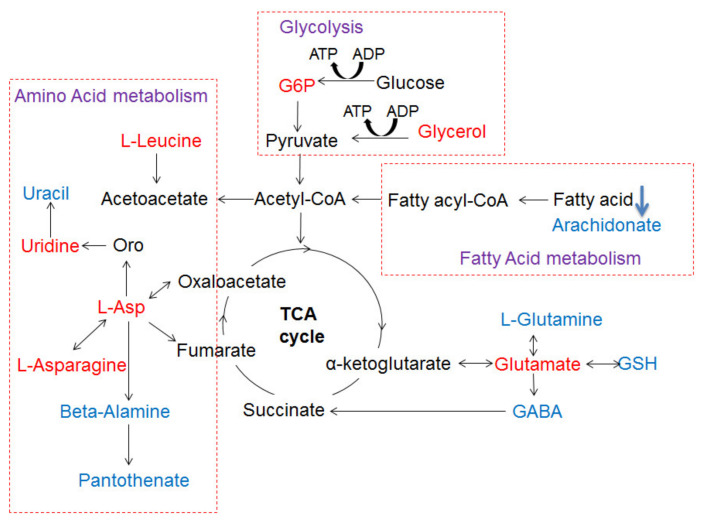
The Schematic sketch of pathway analysis based on differentially produced metabolites identified in liver. Red color and blue color denote up-regulated and down-regulated, respectively.

**Table 1 nanomaterials-11-02656-t001:** Significantly changed metabolites in liver induced by AuNRs or Au@Ag NRs at 2 w. Metabolites were identified by performing PLS-DA analysis and *t* test between nanorod-treated groups and control group (*n* = 3 or 4) using XploreMET software. * *p* < 0.05 and ** *p* < 0.01 identified by Student’s *t* test, comparting to controls, respectively. The fold change (FC) was obtained by comparing relevant metabolites in the nanorod groups with the control group. Corr., correlation coefficients; VIP, variable importance in projection.

Class	Metabolite Name	Au/N			Ag/N		
		FC	Corr.	VIP	FC	Corr.	VIP
Alcohols	Glycerol				7.998 **	0.97	1.8
Amino Acid	Pyroglutamic acid	4.443 *	0.9	2.7	/	/	/
	L-Threonine	1.536 **	0.85	2.6	/	/	/
	L-Asparagine	/	/	/	2.141 *	0.73	1.4
	L-Leucine	/	/	/	4.409 *	0.89	1.7
	Gamma-Aminobutyric acid	/	/	/	0.346 *	−0.9	1.7
	L-Glutamine	/	/	/	0.473	−0.73	1.4
Carbohydrates	L-Arabinose	/	/	/	3.637 **	0.9	1.7
	Cellobiose	/	/	/	19.945 *	0.91	1.7
	Isomaltose	/	/	/	11.056 *	0.84	1.6
	Kojibiose	/	/	/	18.236 **	0.99	1.9
	Alpha-Lactose	/	/	/	1.592 *	0.79	1.5
	D-Maltose	/	/	/	7.746 *	0.89	1.7
	Mannitol	/	/	/	1.958 *	0.86	1.6
	L-Arabitol	/	/	/	0.521 *	−0.8	1.5
	Ribitol	/	/	/	0.521 *	−0.8	1.5
Fatty Acids	Tetracosanoic acid	/	/	/	2.87 *	0.87	1.6
	Arachidonic acid	/	/	/	0.496 **	−0.9	1.7
	Docosahexaenoic acid	/	/	/	0.602 *	−0.79	1.5
	Heptadecanoic acid	/	/	/	0.605 *	−0.87	1.6
	Myristic acid	/	/	/	0.386 *	−0.81	1.5
	Oleic acid	/	/	/	0.563 *	−0.79	1.5
	Palmitic acid	/	/	/	0.627 *	−0.87	1.6
	Palmitoleic acid	/	/	/	0.720 *	−0.83	1.6
Lipids	MG160	/	/	/	0.701 **	−0.88	1.6
	MG182	/	/	/	0.753*	−0.89	1.7
Nucleotide	Uridine	3.213 *	0.78	2.3	8.485 *	0.89	1.7
Organic Acids	Orotic acid	/	/	/	0.064 **	−0.91	1.7
	Petroselinic acid	/	/	/	0.553 **	−0.91	1.7
Vitamin	Pantothenic acid	/	/	/	0.339 **	−0.9	1.7
	Alpha-Tocopherol	/	/	/	0.785 *	−0.83	1.6

## Data Availability

All data in this study will be available from the corresponding author upon reasonable request.

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
