# Peer review of "The Bio-Persistence of Reversible Inflammatory, Histological Changes and Metabolic Profile Alterations in Rat Livers after Silver/Gold Nanorod Administration"

_nanomaterials, 2021, doi:10.3390/nano11102656_

Round 1

Reviewer 1 Report

In the title is indicated silver/gold nanorods and it the discussion authors mainly spoke about silver nanoparticles, but you did not study the effect of silver nanoparticles and of Au@Ag NRs

Abstract need to be rewritten, is less informative

Introduction need to be improved, the novelty of the study is not evident

The concentration of nanoparticles used in experiments was relatively low, why author used this concentration?

In my opinion 4 animals are not enough for experiment

Lines 155-156 why timme of collection in excperiments during two weeks in both experiments is different?

“. In this study, the biological effects of AgNPs intravenously administered to 380 rats were comprehensively assessed” according to the presented data is not true, the authors studied the effect of gold or complex particles and not silver nanoparticles

Reviewer 2 Report

The authors present a study on the long term toxicity of silver-coated gold nanorods in a murine model. They claim this is a novel study, but several other papers on long term toxicity of silver nanoparticles have been published in the last years. I suggest that some of this papers are added as references in the introduction, and the authors should be more specific in describing the novelty of this paper. 

The nanoparticles studied in this paper are silver-coated gold nanorods. This is quite an unconventional choice if the aim was studying the toxicity of nanosilver. This contrast between the title of the paper and the abstract (focused completely on silver) is misleading to the reader. It should be stated more clearly what is the aim of the paper and why the authors choose this particular nanomaterial. In the introducion, the authors write that " It would be much easier to compare total bio-effects of AuNRs with those of AgNPs". I can't understand the meaning of the sentence. 

In the experimental part, paragraph 2.2, no refrence is cited for the synthesis of the nanorods. If this is not a novel synthesis a reference should be added.

I have also some concerns on the results:

paragrpah 3.1: the authors report that Ag has a preferential lateral growth, and this is supported by a greater increase in diameter with respect to the length, but this is contrast with the claim that Ag forms a homogeneous layer of 5 nm thickness. The zeta potential values are referred to the as prepared NR? Is this compatible with similar preparation from the literature? Is there any evidence of the fact that silver is present and localized on the surface of the nanorods? What is the percentage of silver in the Ag-Au nanorods? And in the Au nanordos? The Au nanorods contain silver as well. Is this in a significant percentage? No data are presented on the toxicity of this silver. 

I think that this paragraph should be improved with more characterization.

paragraph 32.:  "the blood content of Au was a little bit 230 more after Au@Ag NRs exposure" judging from figure 2 the content is the same, this claim should be better supported. Moreover, it should be stated if the injected quantity of Au is the same or not (or did the authors work with a similar total metal concentration, i.e. Ag+Au?) 

The paper also needs some editing, since several typos are present, for example:

line 44: "The Due" seems to be a typo

line 102: is 1 a reference or a typo? In the latter case a reference should be added for the synthetic procedure

line 206, line 26: "the text continues here" should be removed

line 375: "autho" must be removed

The study on toxicity, however, is deep and scientifically sound, and can be of interest for the researchers in the field. I therefore recommend that the manuscript is edited and improved in the parts that I mentioned, before it can be published in this journal.

Round 2

Reviewer 1 Report

Dear Authors, 

Thank you for taking into account my comments.

Reviewer 2 Report

The authors ansewred to all my doubts on the paper and they significantly improved the quality of the manuscript. The paper is now acceptable in its present form.